# Repurposable Drugs That Interact with Steroid Responsive Gene Targets for Inner Ear Disease

**DOI:** 10.3390/biom12111641

**Published:** 2022-11-05

**Authors:** Alexander A. Missner, James Dixon Johns, Shoujun Gu, Michael Hoa

**Affiliations:** 1Georgetown University School of Medicine, Washington, DC 20007, USA; 2Department of Otolaryngology-Head and Neck Surgery, Georgetown University Medical Center, Washington, DC 20007, USA; 3Auditory Development and Restoration Program, National Institute on Deafness and Other Communication Disorders, National Institutes of Health, Bethesda, MD 20892, USA

**Keywords:** corticosteroids, transtympanic steroids, sudden sensorineural hearing loss, Meniere’s disease, autoimmune inner ear disease, drug repurposing, cochlea, transcriptome, RNA sequencing, spiral ganglion neurons, stria vascularis

## Abstract

Corticosteroids, oral or transtympanic, remain the mainstay for inner ear diseases characterized by hearing fluctuation or sudden changes in hearing, including sudden sensorineural hearing loss (SSNHL), Meniere’s disease (MD), and autoimmune inner ear disease (AIED). Despite their use across these diseases, the rate of complete recovery remains low, and results across the literature demonstrates significant heterogeneity with respect to the effect of corticosteroids, suggesting a need to identify more efficacious treatment options. Previously, our group has cross-referenced steroid-responsive genes in the cochlea with published single-cell and single-nucleus transcriptome datasets to demonstrate that steroid-responsive differentially regulated genes are expressed in spiral ganglion neurons (SGN) and stria vascularis (SV) cell types. These differentially regulated genes represent potential druggable gene targets. We utilized multiple gene target databases (DrugBank, Pharos, and LINCS) to identify orally administered, FDA approved medications that potentially target these genes. We identified 42 candidate drugs that have been shown to interact with these genes, with an emphasis on safety profile, and tolerability. This study utilizes multiple databases to identify drugs that can target a number of druggable genes in otologic disorders that are commonly treated with steroids, providing a basis for establishing novel repurposing treatment trials.

## 1. Introduction

Corticosteroids, oral or transtympanic, may be used for the treatment of inner ear diseases such as sudden sensorineural hearing loss (SSNHL), Meniere’s disease (MD), and autoimmune inner ear disease (AIED). The mechanisms of these diseases are all poorly understood and there is a lack of proven treatment options that consistently reduce disease burden and improve symptoms [1,2,3]. Due to the lack of efficacious interventions, steroids are often utilized in the management of these diseases. Although the data remains inconsistent, there have been studies to support the use of steroids in many of these disorders [1,2,3,4,5]. The use and reliance of steroids in these otologic disorders necessitates research on how steroids effect the inner ear, which is currently not well characterized. One option to elucidate these steroid-mediated effects is to examine how pharmacologically modulating genetic targets of steroids impacts these diseases that are difficult to study. This approach is treatment focused, addresses the gap in treatment options, generates data for future study, and is a novel way to explore how steroids may be targeting these disorders.

SSNHL, MD, and AIED are potentially devastating disorders that affect the inner ear. The pathogenesis and mechanisms underlying these disorders remain poorly understood. SSNHL is defined as rapid-onset sensation of hearing impairment in one or both ears, involving a decrease in hearing of ≥30 decibels affecting at least 3 consecutive frequencies, within a 72-h period [1]. Accompanying symptoms often include tinnitus, dizziness and/ or vertigo in 30–60% of patients [1]. It is a primarily idiopathic disease with 10% of cases linked to identifiable etiologies including viral, bacterial, tumors, thrombosis, and immunologic. SSNHL is believed to result from abnormalities of the cochlea, auditory nerve, or processes of central auditory perception causing sudden unilateral or bilateral hearing loss [1].

AIED is an immune-mediated inner ear disease that often results in progressive SNHL. In AIED, patients experience frequent episodes of hearing loss, which may occur bilaterally [3,6]. AIED has historically been diagnosed based on whether patients respond to steroids [3,7]. In AIED, steroids have been shown to be largely effective in hearing improvement, although patients may develop steroid resistance over time [8]. Similar to SSNHL, there remains a lack of proven treatments for steroid-refractory patients [6].

MD is defined by spontaneous vertigo attacks accompanied by sensorineural hearing loss (SNHL) that can occur during, before, or after these debilitating attacks, and fluctuating aural symptoms, such as ear fullness and tinnitus [2]. Diagnosis requires two or more spontaneous attacks of vertigo, each lasting 20 min to 12 h; audiometrically documented low- to mid-frequency SNHL in affected ear on at least one occasion before, during, or after one episodes of vertigo; fluctuating aural symptoms in affected ear; and other causes excluded [2].

The combined prevalence of these disorders is estimated to be 210 to 232 per 100,000 people in the United States [2,9,10,11,12]. In addition, chronic hearing loss and lack of reliable treatments is a source of disability and morbidity for the disorders [13,14,15,16,17]. Similarly, studies assessing quality of life in MD patients have found profound levels of psychological and economic handicap [18,19,20,21,22]. Across these disorders, steroids are one of few options that are routinely used, and in SSNHL and MD, they are recommended by established clinical practice guidelines despite significant heterogeneity in patient responsiveness in these disorders [1,2]. Therefore, it is of great interest to better characterize the role of steroids in these disease processes.

There is a need to investigate drugs in these otologic disorders in order to further understand disease mechanisms and develop reliable therapeutics that lead to higher rates of recovery. Since steroids are one of few medications used to treat these disorders, assessing the genes they interact with is a method to test therapeutics for these diseases. In a previous study, we identified inner ear-expressed steroid-responsive genes that could be potentially targeted [23]. Through cross-referencing these steroid responsive genes in the cochlea [24] with published single-cell and single-nucleus transcriptome datasets, we have shown that steroid-responsive, differentially regulated genes are expressed in spiral ganglion neurons (SGN) and stria vascularis (SV) cell types. These genes have been associated with anti-inflammatory, cell stress response, and apoptosis pathway signaling [23,24]. Given the use of steroids in these disease processes, it may be inferred that these differentially regulated gene targets may represent potential druggable targets.

Identifying FDA-approved medications that directly interact with these gene targets (“druggable”) may provide an opportunity for repurposing these drugs for the treatment of inner ear disorders in addition to gaining an understanding of gene targets that may play a role in these disorders. In light of these steroid responsive genes, this study identifies druggable gene targets and their respective potentially repurposable FDA-approved therapeutics and reviews evidence associated with safety profile and tolerability. Further research is needed to elucidate the effectiveness of these therapies. The identification of these potential therapeutics may establish a foundation for the investigation of novel treatment options for these disorders.

## 2. Materials and Methods

### 2.1. Identification of Steroid-Responsive Genes in the Mammalian Spiral Ganglion Neurons and Stria Vascularis Cell Types

Single-cell and single-nucleus RNA transcriptome datasets from wildtype adult mice were previously cross-referenced with microarray data from whole isolated mouse cochleae. These mice were treated with systemic steroids (SS), transtympanic steroids (TTS), or saline as the control, and are described in Trune et al., 2019 [23,24]. In summary, data presented are from normal BALB/cJ mice at 2 to 3 months old treated systematically or transtympanically with 0.7 mg/kg of dexamethasone sodium phosphate and 10 mg/kg prednisolone. Systematic treatment involved delivery (200 μL) of dexamethasone given as a single subcutaneous injection or prednisolone as a single oral dose. Transtympanic injection of these steroids involved 5 μL into both middle ears through the tympanic membranes. We utilized data from 6 mice treated with TTS prednisolone, 7 mice treated with TTS dexamethasone, 8 control (saline treated) mice, 8 mice treated with oral prednisolone, and 7 mice treated with subcutaneous dexamethasone. Differentially expressed genes (DEGs) were identified in two treatment comparisons in adult mouse cochleae: (1) SS compared to control, and (2) TTS compared to control. Briefly, criteria for determining DEGs included a false-discovery rate (FDR) adjusted *p* < 0.05. These previous analyses identified several lists of differentially expressed steroid-responsive genes and were cross-referenced with published single-cell RNA-seq dataset from P25–P27 mouse spiral ganglion neurons (SGN) and single-nucleus RNA-seq dataset from P30 stria vascularis (SV) in the wild-type mammalian cochlea to derive lists of steroid-responsive genes expressed for the two treatment comparisons in SGN and SV cell types [24,25,26]. We utilized these previously derived lists of DEGs for our subsequent analyses, as displayed in Appendix A in Nelson et al., 2022 [23,24,25,27].

### 2.2. Identification of Druggable Gene Targets Amongst Spiral Ganglion Neurons and Stria Vascularis Cell Types

The previously generated DEGs from the treatment comparisons (SS vs. control, and TTS vs. control) with expression in either SGN and SV cell types were then cross-referenced with DrugBank (https://go.drugbank.com/pharmaco/transcriptomics) (accessed on 7 December 2021) [28], Pharos (https://pharos.ncats.nih.gov/) (accessed on 7 December 2021) [29], and the LINCS Repurposing app (https://clue.io/repurposing-app) (accessed on 7 December 2021) [30] to identify FDA-approved drugs that may interact with these highly expressed genes within the SGN and SV. Pharos is a web interface for the Target Central Resource Database (TCRD) created by the “Illuminating the Druggable Genome” program, which collates known drug-protein interactions, including both FDA-approved drugs and small molecules [29,31] that our group has previously utilized to identify druggable gene targets and their respective FDA-approved medications in the setting of cisplatin treatment [32]. Similarly, both DrugBank and LINCS have been utilized to identify potential druggable gene targets and repurposable therapeutics [33,34,35,36]. Thus, based on these databases, the lists of druggable DEGs in the two treatment comparisons that are expressed in SGN cells and SV cell types (Appendix A) are provided.

Genes that demonstrated a drug-gene log expression value greater than 1.20 (representing a fold change greater than 2.3) were included to evaluate the top 75 druggable gene targets according to levels of differential genetic expression. Drugs approved by the FDA were selected from DrugBank, Pharos, and LINCS, and non-FDA approved drugs were not included in our analysis. The remaining drugs were screened for safety data, adverse events, contraindications, and black-box warnings based on FDA safety labels. The mechanism of action (MOA) of the drugs given by these databases are reported in the results section. The MOA “inhibitor” is given to drugs that decrease gene expression (blocker, desensitizer, negative allosteric modulator, and antagonist); “activator” is assigned to drugs that increase gene expression (agonist); and “ligand”/“unknown” drugs either have a mixed effect or are unknown according to DrugBank, Pharos, and LINCS.

To further refine the list of candidate drugs with repurposing potential, orally administered medications were isolated and evaluated for safety. Given our previous work identifying strong expression of steroid-responsive gene targets amongst spiral ganglion neurons [23], we then chose to focus on orally administered, FDA approved drugs known to interact with genes strongly expressed in the SGN as well as in SV cell types. These genes included *Grin1*, *Pdxk*, *Kcnh2*, *Kcna5*, *Kcnq2*, *Txnrd1*, *Nfkbia*, *Per1*, *Cacna1a*, *Nrxn1*, *Atf3*, *Sbk1*, *Mast1*, *Sphk1*, *Kcnt1*, *Fdx1*, *Fkbp5*, *Csn1kd*, *Nisch*, and *Tufm*. Following identification of candidate drug targets, the safety profiles were assessed. Exclusion criteria included drugs with significant adverse event profiles including, abuse potential, neurologic side effects, severe hypersensitivity, cardiac risks (arrythmia, QT prolongation, cardiac arrest), hematologic adverse events, immunosuppressants, severe hepatotoxic agents, ototoxic drugs, and drugs removed from the US market (bepridil and cisapride).

The application Drugmonizome through their “Drug Set Enrichment Analysis” (DSEA) (https://maayanlab.cloud/drugmonizome/#/DrugSetEnrichment/Overlap) (accessed on 24 April 2022) was used to identify side effects of the drug classes selected for further analysis through the SIDER tool [37,38]. Their algorithm extracts side effects from drug label package inserts from national registries, and the frequencies of drug reactions from tables and free-text [38]. We display the ten most prevalent side effects for drugs classes of interest below based on *p*-values for statistical significance of the frequencies of these adverse events (AEs) from SIDER. This SIDER database contains adverse drug reactions for a large number of drugs and is a proven tool for use in drug repurposing analysis [39,40].

## 3. Results

### 3.1. Overview

The orally administered FDA approved drugs that interact with steroid-responsive genes expressed in the SGN and SV were placed into categories below. Table 1, Table 2, Table 3, Table 4, Table 5, Table 6, Table 7, Table 8, Table 9 and Table 10 contain these medications and whether the corresponding gene expression changed (up or downregulated) when treated according to different steroid administration routes (oral or TTS) versus control (saline). In the tables, Y (yes, enriched) conveys that the gene displays differential expression according to administration route in the cochlea, based on microarray data from the whole cochlea. NC (no change) means that there is no change in cochlea gene expression after steroid treatment. All genes listed are expressed in the SGN and /or SV, based on previously published single-cell and single-nucleus data [23]. For example, if a “SGN” column shows Y for a gene-drug interaction for TTS injection (compared to control) but NC for systemic steroid (compared to control), TTS administration in the cochlea resulted in enriched expression of the given gene when compared to the saline treated cochlea, and the gene is expressed in the SGN. The oral corticosteroid, however, did not result in differential expression of the gene in the cochlea despite the gene being expressed in the SGN. In the following section, we have identified potentially repurposable drugs that may act on druggable (steroid-responsive) gene targets in both the SGN and SV of the adult mammalian cochlea based on gene-drug interactions.

### 3.2. FDA-Approved Drugs with Repurposing Potential for Sudden Sensorineural Hearing Loss Organized by Class

#### 3.2.1. Antidepressants

Table 1 shows the drugs that fall under the category of antidepressants with several mechanisms of action involving neurotransmitters. The genes that these drugs interact with are highly expressed in both the SGN and the SV indicated by their high differential expression levels after oral and TTS treatment (Table 1). These drugs penetrate the blood–brain barrier (BBB) [41,42,43,44,45,46]. They are widely used for the treatment of major depressive disorder [47]. These medications are additionally used in other scenarios such anxiety, obsessive compulsive disorder, and posttraumatic stress disorder [48,49,50]. Although these antidepressants have FDA black box warnings for suicidality, they are highly prescribed and well tolerated among the majority of patients [51,52]. For potential use of these drugs in otologic disorders, these drugs would be tested as a short-term treatment as well, which may mitigate these adverse events. Tricyclic antidepressants (TCAs) can cause cardiac toxicity, including prolonging electrocardiogram measures, and therefore must be carefully monitored in patients [53]. They are included in this analysis due to their widespread use and tolerability for a variety of non-depressive disorders such as migraine, neuropathic pain, vestibular conditions, and recently in a number of case studies of patients with SSNHL [54,55,56]. Figure 1 shows the shared side effect profiles of these antidepressants grouped by the frequency in which they occur.

In recent drug-safety trials for use in COVID-19, fluoxetine and fluvoxamine were tested for a potential reduction of symptom severity, and were shown to be safely tolerated in the patient cohort [57,58]. The proposed mechanisms that motivated these trials were due to SSRI reducing levels of several proinflammatory cytokines.

**Table 1 biomolecules-12-01641-t001:** FDA approved, orally administered antidepressants that interact with steroid-responsive genes in the cochlea localized to SGN and SV cell types.

Gene	Drug	Drug Class	MOA	SGN: SS > Control	SGN: TTS > Control	SV: SS > Control	SV: TTS > Control
Kcnh2	Amitriptyline	TCA	Inhibitor	Y	Y	Y	Y
Kcnh2	Doxepin	TCA	Inhibitor	Y	Y	Y	Y
Kcnh2	Fluoxetine	SSRI	Inhibitor	Y	Y	Y	Y
Kcnh2	Fluvoxamine	SSRI	Unknown	Y	Y	Y	Y
Kcnh2	Imipramine	TCA	Inhibitor	Y	Y	Y	Y
Grin1	Milnacipran	SSNRI	Inhibitor	NC	Y	NC	Y
Kcnh2	Nefazodone	SARI	Inhibitor	Y	Y	Y	Y

Enriched (Y), no change in expression (NC), mechanism of action (MOA), spiral ganglion neurons (SGN), stria vascularis (SV), oral systemic steroid (SS), transtympanic steroid (TTS), saline control (control), serotonin specific reuptake inhibitor (SSRI), selective serotonin and norepinephrine reuptake inhibitor (SSNRI), tricyclic antidepressant (TCA), serotonin antagonist and reuptake inhibitor (SARI).

#### 3.2.2. Antipsychotics

The antipsychotics that interact strongly with steroid-responsive genes highly expressed in the SGN and SV are shown in Table 2. These agents’ AEs are displayed in Figure 2. These drugs interact with the dopamine receptors in the central nervous system to improve the symptoms of psychosis [59]. Each of the three drugs in this category has a different profile in terms of the interaction with the dopamine receptor. First generation ‘typical’ antipsychotics (FGA) such as thioridazine and loxapine are an older class of antipsychotic than second generation ‘atypical’ antipsychotics. The FGAs are effective at reducing positive symptoms associated with schizophrenia, but are largely limited by extrapyramidal motor side effects (EPS) (Figure 2: choreoathetosis, akinesia, tardive dyskinesia, and muscle rigidity), galactorrhea (Figure 2) and cognitive dulling [59]. These adverse effects are likely mediated by high dopamine D_2_ receptor antagonism and occupancy.

The atypical antipsychotics are considered second-generation antipsychotics (SGA). Serotonin 5-HT_2A_ receptor antagonism in combination with D_2_ receptor antagonism is thought to be the hallmark pharmacology of the SGAs. The SGAs have reduced EPS liability compared to the FGAs, but can be associated with increased weight gain and metabolic burden mediated by unintended off-target pharmacological interactions [59]. Aripiprazole is an SGA with D_2_ receptor partial agonist effects rather than D_2_ antagonism. Presynaptic partial agonism at D_2_ receptors has allowed for a further reduction in EPS and hyperprolactinemia, but postsynaptic partial agonism at D_2_ receptors has been associated with relatively high levels of akathisia (Figure 2) and difficulty with addiction [59]. The antipsychotics generally carry a black box warning that they increase mortality in the elderly with dementia [60]. However, these medications are commonly used to treat psychosis and schizophrenia. While tardive dyskinesia (Figure 2) may develop with long term use, in the setting of otologic diseases, the treatment protocol is expected to be time limited.

**Table 2 biomolecules-12-01641-t002:** FDA approved, orally administered antipsychotics that interact with steroid-responsive genes in the cochlea localized to SGN and SV cell types.

Gene	Drug	MOA	SGN: SS > Control	SGN: TTS > Control	SV: SS > Control	SV: TTS > Control
Grin1	Aripiprazole	Ligand	NC	Y	NC	Y
Kcnh2	Thioridazine	Inhibitor	Y	Y	Y	Y
Kcnt1	Loxapine	Activator	NC	Y	NC	Y

Enriched (Y), no change in expression (NC), mechanism of action (MOA), spiral ganglion neurons (SGN), stria vascularis (SV), oral systemic steroid (SS), transtympanic steroid (TTS), saline control (control).

#### 3.2.3. Central Nervous System: Other

Drugs known to have efficacy in the central nervous system that interact strongly with genes highly expressed in the SGN are shown in Table 3. These genes were all enriched with TTS administration. These drugs have actions on the neurotransmitter systems of the central nervous system but there is no one unifying mechanism. These drugs are used in a variety of disorders such as neuropathic pain (gabapentin), Alzheimer’s disease (donezipil and memantine), seizures (phenytoin), and dopamine modulation (amantadine and atomoxetine). These drugs have side effect profiles that cause drug-induced movement disorders (hypokinesia, hemiplasia), and psychiatric side effects (excoriation, psychiatric disorder, personality disorder, and other listed effects in Figure 3) [61,62]. Many of these side effects are attributed to clinical features or progression of the disorders that they treat, thus making these side effects unlikely in short term treatment of inner ear diseases, particularly in patients without CNS comorbidities. These drugs pass the BBB, which is of value in considering drugs that may be used to treat ear pathologies.

**Table 3 biomolecules-12-01641-t003:** FDA approved, orally administered CNS drugs that interact with steroid-responsive genes in the cochlea localized to SGN and SV cell types.

Gene	Drug	MOA	SGN: SS > Control	SGN: TTS > Control	SV: SS > Control	SV: TTS > Control
Grin1	Donepezil	Inhibitor	NC	Y	NC	Y
Cacna1a	Gabapentin	Inhibitor	NC	Y	NC	Y
Grin1	Gabapentin	Inhibitor	NC	Y	NC	Y
Grin1	Memantine	Inhibitor	NC	Y	NC	Y
Kcnh2	Phenytoin	Inhibitor	Y	Y	Y	Y
Grin1	Amantadine	Inhibitor	NC	Y	NC	Y
Grin1	Atomoxetine	Inhibitor	NC	Y	NC	Y

Enriched (Y), no change in expression (NC), mechanism of action (MOA), spiral ganglion neurons (SGN), stria vascularis (SV), oral systemic steroid (SS), transtympanic steroid (TTS), saline control (control).

#### 3.2.4. Over the Counter

Over the counter (OTC) drugs identified that interact with steroid-responsive genes in the SGN and SV (Table 4) are particularly well-tolerated with low side effect profiles. The most frequent SIDER effects for these drugs involve application site papules and abscesses for diclofenac, although in the case of inner ear disease, we are proposing oral administration. Rare poisoning with aspirin can include severe metabolic derangements and significant toxicity [63]. Overall, OTC drugs are well tolerated with rare AEs that terminate after cessation of treatment, making them of therapeutic interest for repurposing.

**Table 4 biomolecules-12-01641-t004:** FDA approved, orally administered over-the-counter (OTC) drugs that interact with steroid-responsive genes in the cochlea localized to SGN and SV cell types.

Gene	Drug	MOA	SGN: SS > Control	SGN: TTS > Control	SV: SS > Control	SV: TTS > Control
Grin1	Acetylcysteine	Activator	NC	Y	NC	Y
Nfkbia	Aspirin	Inhibitor	NC	Y	NC	Y
Kcnq2	Diclofenac	Inhibitor	NC	Y	NC	Y
Grin1	Guaifenesin	Inhibitor	NC	Y	NC	Y
Grin1	Magnesium carbonate	Inhibitor	NC	Y	NC	Y
Tufm	Zinc	Unknown	Y	Y	Y	Y
Tufm	Zinc acetate	Unknown	Y	NC	Y	Y
Kcnh2	Pentoxyverine	Inhibitor	Y	Y	Y	Y
Pdxk	Pyridoxine	Ligand	NC	Y	NC	NC

Enriched (Y), not enriched (NC), mechanism of action (MOA), spiral ganglion neurons (SGN), stria vascularis (SV), oral systemic steroid (SS), transtympanic steroid (TTS), saline control (control).

#### 3.2.5. Adrenergic Modulators

Table 5 shows alpha receptor modulators (doxazosin, prazosin, terazosin, and tizanidine, and ephedrine) and a combined alpha and beta blocker (carvedilol) that interact with steroid responsive genes in the SGN and SV. These drugs are highly prescribed agents with well-characterized side effect profiles. As seen in Figure 4, a rare effect of alpha blockers is floppy iris syndrome which is a complication of cataract surgery in patients receiving alpha-1 antagonists [64]. The other AEs shown are well characterized due to beta or alpha receptor mediated modulations. These effects terminate when the drug is ceased. Tizanidine is a muscle relaxant used to treat muscle spasticity, spinal cord injuries, or multiple sclerosis. It is widely prescribed, with AEs of dry mouth, sleepiness, weakness, and dizziness [65]. More serious AEs include CNS effects such as hallucinations, and sensory effects such as spasticity and tingling, although these are rarely reported [65].

**Table 5 biomolecules-12-01641-t005:** FDA approved, orally administered alpha and beta blockers that interact with steroid-responsive genes in the cochlea localized to SGN and SV cell types.

Gene	Drug	MOA	SGN: SS > Control	SGN: TTS > Control	SV: SS > Control	SV: TTS > Control
Kcnh2	Carvedilol	Inhibitor	Y	Y	Y	Y
Kcnh2	Doxazosin	Inhibitor	Y	Y	Y	Y
Kcnh2	Prazosin	Inhibitor	Y	Y	Y	Y
Kcnh2	Terazosin	Inhibitor	Y	Y	Y	Y
Atf3	Ephedrine-(racemic)	Activator	NC	Y	NC	Y
Nisch	Tizanidine	Activator	NC	Y	NC	Y

Enriched (Y), no change in expression (NC), mechanism of action (MOA), spiral ganglion neurons (SGN), stria vascularis (SV), oral systemic steroid (SS), transtympanic steroid (TTS), saline control (control).

#### 3.2.6. Calcium Channel Blockers

Table 6 contains calcium channel blockers which reduce calcium activity in the heart and arteries. They are used to treat hypertension, angina, and certain arrythmias. They are highly prescribed and well tolerated, but have potential for cardiac toxicity and muscle fatigue (Figure 5). Thus, these drugs must be dosed carefully in studies to evaluate their effect in inner ear disease to minimize potential toxicity.

**Table 6 biomolecules-12-01641-t006:** FDA approved, orally administered calcium channel blockers that interact with steroid-responsive genes in the cochlea localized to SGN and SV cell types.

Gene	Drug	MOA	SGN: SS > Control	SGN: TTS > Control	SV: SS > Control	SV: TTS > Control
Kcna5	Nifedipine	Inhibitor	NC	Y	NC	Y
Cacna1a	Verapamil	Inhibitor	NC	Y	Y	Y
Kcnh2	Verapamil	Inhibitor	Y	Y	Y	Y

Enriched (Y), no change in expression (NC), mechanism of action (MOA), spiral ganglion neurons (SGN), stria vascularis (SV), oral systemic steroid (SS), transtympanic steroid (TTS), saline control (control).

#### 3.2.7. Ion Modulators

Ion modulators are listed in Table 7, including calcium supplementation (calcium citrate and calcium phosphate) in addition to spironolactone, a potassium-sparing diuretic. Calcium supplementation is given for a range of conditions such as calcium deficiency, vitamin D deficiency, tetany, and associated conditions. The major side effects of calcium supplementation (Figure 6) include sensation of heat waves (sense of oppression), vasodilation, gastrointestinal irritation, kidney stones, and cardiovascular disease with doses over 1000 mg/d such as increased risk of myocardial infarction [66,67]. Thus, for potential repurposing in the treatment of inner ear diseases such as MD, SSNHL, and AIED, dosage would need to fall below this threshold.

Spironolactone is a mineralocorticoid receptor antagonist used as a potassium-sparing diuretic to treat hypokalemia, hyperaldosteronism, and dermatologic conditions associated with androgens [68]. It is associated with AEs such as electrolyte abnormalities, hyperkalemia, nausea, vomiting, headache, and antiandrogenic symptoms such as galactorrhea [69,70].

**Table 7 biomolecules-12-01641-t007:** FDA approved, orally administered drugs that modulate ions including calcium channel blockers and spironolactone that interact with steroid-responsive genes in the cochlea localized to SGN and SV cell types.

Gene	Drug	MOA	SGN: SS > Control	SGN: TTS > Control	SV: SS > Control	SV: TTS > Control
Nrxn1	Calcium citrate	Activator	NC	Y	NC	Y
Nrxn1	Calcium phosphate	Activator	NC	Y	NC	NC
Cacna1a	Spironolactone	Inhibitor	NC	Y	NC	Y

Enriched (Y), no change in expression (NC), mechanism of action (MOA), spiral ganglion neurons (SGN), stria vascularis (SV), oral systemic steroid (SS), transtympanic steroid (TTS), saline control (control).

#### 3.2.8. Antimicrobials

Table 8 displays antifungals (isavuconazole) and an antibiotic (erythromycin) that interact with genes highly expressed in the SGN and SV through oral and TTS treatment. Isavuconazole is well tolerated and does not seem to increase cardiac risk, unlike other antifungals in the triazole class [71]. Isavuconazole has a favorable safety profile, with mild hepatotoxicity and drug interactions [72]. Erythromycin is a widely used antibiotic for respiratory tract, genital, and other infections, with a proven safety profile [73]. Common AEs include vomiting, abdominal cramps, and diarrhea, and rarer serious AEs include hepatotoxicity, allergies, and cardiac risks [74]. Nonetheless, it is widely prescribed and typically well tolerated.

**Table 8 biomolecules-12-01641-t008:** FDA approved, orally administered antifungal and antibiotics that interact with steroid-responsive genes in the cochlea localized to SGN and SV cell types.

Gene	Drug	MOA	SGN: SS > Control	SGN: TTS > Control	SV: SS > Control	SV: TTS > Control
Kcna5	Isavuconazole	Inhibitor	NC	Y	NC	Y
Kcnh2	Isavuconazole	Inhibitor	Y	Y	Y	Y
Kcnh2	Erythromycin	Inhibitor	Y	Y	Y	Y

Enriched (Y), no change in expression (NC), mechanism of action (MOA), spiral ganglion neurons (SGN), stria vascularis (SV), oral systemic steroid (SS), transtympanic steroid (TTS), saline control (control).

#### 3.2.9. Antihistamines

Table 9 includes three widely prescribed antihistamines. Hydroxyzine is a first-generation antihistamine that can readily cross the BBB and is used to treat itchiness, anxiety, nausea, and motion sickness [75,76]. It can cause sedation, dizziness, hypotension, headaches, and tinnitus. Rarely, it can cause more severe CNS effects like hallucinations. Its ability to cross the BBB may be of value for use in the treatment of some inner ear diseases. Recently, hydroxyzine was evaluated for potential use in COVID-19 trials for its safety profile that researchers believed may reduce disease burden [77]. Orphenadrine is an H1 antihistamine used as a muscle relaxant to help with motor control in Parkinson’s disease [78]. It also has anticholinergic activity causing AEs such as dry mouth, dizziness, drowsiness, constipation, urine retention, blurred vision, and headache [78]. Orphenadrine is contraindicated in patients with digestive conditions, bladder and prostate disorders, and myasthenia gravis. Loratadine is a second-generation antihistamine that penetrates the BBB less than the first-generation drugs, and is more selective for the H1 receptor, giving it less sedative and autonomic AEs [76,79].

**Table 9 biomolecules-12-01641-t009:** FDA approved, orally administered antihistamines that interact with steroid-responsive genes in the cochlea localized to SGN and SV cell types.

Gene	Drug	MOA	SGN: SS > Control	SGN: TTS > Control	SV: SS > Control	SV: TTS > Control
Kcnh2	Loratadine	Inhibitor	Y	Y	Y	Y
Kcnh2	Hydroxyzine	Inhibitor	Y	Y	Y	Y
Grin1	Orphenadrine	Inhibitor	NC	Y	Y	Y

Enriched (Y), no change in expression (NC), mechanism of action (MOA), spiral ganglion neurons (SGN), stria vascularis (SV), oral systemic steroid (SS), transtympanic steroid (TTS), saline control (control).

#### 3.2.10. Fostamatinib

Fostamatinib, as seen in Table 10, is a tyrosine kinase inhibitor used to treat chronic immune thrombocytopenia [80,81]. It modifies expression of the genes that are strongly expressed in the inner ear (Table 10). This makes it of therapeutic interest for these inner ear diseases. Fostamatinib is a first-class medication to reduces immune-mediated platelet destruction. It is well tolerated with AEs in the following classes: gastrointestinal (diarrhea, nausea, abdominal pain) as well as hypertension, respiratory infection, and decreased white blood cell count [80]. In the major clinical trials that led to FDA approval of the drug, AEs only led to drug cessation in three patients out of 101 representing less than 3% [81]. It is also currently involved in clinical trials for a variety of autoimmune and neoplastic conditions. Similarly, it is in clinical trials for severe COVID-19 symptoms [82].

**Table 10 biomolecules-12-01641-t010:** Fostamatinib, an FDA approved, orally administered drug interacts with steroid-responsive genes in the cochlea localized to SGN and SV cell types.

Gene	Drug	MOA	SGN: SS > Control	SGN: TTS > Control	SV: SS > Control	SV: TTS > Control
Mast1	Fostamatinib	Inhibitor	NC	Y	NC	Y
Sbk1	Fostamatinib	Inhibitor	NC	Y	NC	Y

Enriched (Y), not enriched (NC), mechanism of action (MOA), spiral ganglion neurons (SGN), stria vascularis (SV), oral systemic steroid (SS), transtympanic steroid (TTS), saline control (control).

## 4. Discussion

### 4.1. Overview

SSNHL, MD, and AIED represent poorly understood and potentially devastating otologic disorders. Although steroids, both systemic or transtympanic, have been shown to potentially provide therapeutic benefit to some patients with these diseases, there is a dearth of alternative therapies in patients that do not respond adequately to steroids. Non-responsiveness to steroids in the setting of these inner ear diseases may be due to several factors, including, but not limited to, poor understanding of underlying disease mechanisms, mixed and potentially opposing intracellular effects of steroids on different genes and proteins, failure to initiate treatment during critical time window, down-regulation of responsive mechanisms over time, and underlying patient factors. Although steroids display inconsistency across the literature with regard to their therapeutic success in these disorders, they are frequently recommended by treatment guidelines [1,2,3,4,5].

In SSNHL, clinical guidelines for treatment primarily involve corticosteroid treatment, either orally or through transtympanic injections (TTS), with prompt treatment within the first two weeks thought to be critical for recovery [1,10,83,84,85]. Oral corticosteroids are indicated within 2 weeks of symptom onset (Grade C evidence) and TTS are used for patients that may not tolerate oral steroid administration or those with incomplete restoration of hearing levels 2–6 weeks from symptom onset (Grade B evidence) [1]. Although clinical trials have not shown one method to be superior, TTS has been shown to demonstrate a lower side effect profile and may have greater inner ear penetration with studies suggesting higher perilymphatic steroid concentrations versus systemic steroid administration [1,86,87,88]. While the mechanisms by which corticosteroids may alleviate SSNHL are unknown, evidence suggests a role in the reduction of inflammation and edema and in the regulation of ion homeostasis [89,90,91,92,93,94].

Despite the use of either oral or transtympanic corticosteroids, the rate of complete recovery remains low and results across the literature demonstrate significant heterogeneity with respect to the effect of corticosteroids, suggesting a need to identify more efficacious treatment options [95,96]. Specifically, a Cochrane review that analyzed 3 randomized control trials (RCTs) comparing systemic steroids vs. placebo for SSNHL, found the role of steroids to be unclear as all trials were shown to have a high risk of bias, with 1 out of the 3 RCTs showing an improvement in 61% of patients in the steroid treated group compared to 32% in the controls [1,92,97,98], while others have not shown better efficacy than placebo. Studies that investigated the use of TTS vs. placebo have similarly shown mixed, inconsistent results. Altogether, clinical guidelines recommend the use of steroids in SSNHL because it is one of few treatment options with any data showing efficacy [1]. Understanding the mechanisms by which corticosteroids act in the cochlea is critical to developing improved targeted therapies and may elucidate the mechanisms of heterogeneity in steroid-responsiveness among patients with SSNHL [1].

In MD clinical practice guidelines, TTS are a recommended option for patients unresponsive to non-invasive treatments [2]. TTS have been shown to reduce vertigo [99,100], they may salvage hearing after an acute MD attack [101,102,103], and may improve tinnitus in MD [104]. Oral steroids are also routinely used in MD, and although robust data is lacking, studies have shown improvement in vertigo [105]. Therefore, MD practice guidelines conclude that IT steroid therapy is well tolerated and a useful option in the treatment of patients with MD, although similar to the other otologic disorders presented, heterogeneity of these results are demonstrated across studies.

Previous studies have identified multiple genes within the SGN and SV cell types that have been implicated as possible targets in the response to treatment of steroid-responsive otologic disorders [23,106]. These target genes in the SGN and SV have been associated with anti-inflammatory pathways including cell stress response and apoptotic pathways, highlighting potential mechanisms for steroid-responsiveness [23,93,94]. Similarly, there is evidence of inflammation and immune dysregulation in inner ear diseases treated with steroids, namely AIED [3].

Through our analysis, we have identified 42 non-steroid, FDA-approved therapeutics with relatively high safety profiles that have been shown to interact with steroid-responsive genes within the SGN and SV cell types. These drugs should be considered for further investigation into their effect in steroid-responsive otologic disease. These medications may represent opportunities to investigate novel therapies in the setting of incomplete or non-response to steroid therapy in the setting of otologic diseases treated with corticosteroids including SSNHL, MD, or AIED.

### 4.2. Repurposing Studies as an Approach Forward

COVID-19 led to a desperate need to quickly identify therapeutics to reduce severity of the novel virus, which prompted an increase in repurposable drug studies, where transcriptome datasets from both mouse models and humans were used to identify potentially repurposable therapies. These studies allow researchers to efficiently utilize gene-drug interactions to explore pathways involved in disease pathologies, identify druggable targets, and utilize bioinformatics tools with increasingly sophisticated algorithms and software. For example, a similar transcriptome-guided analysis was recently published to identify repurposing drugs to treat age-related hearing loss [107], and other studies have utilized this approach for a number of diseases such as cancers and COVID-19 [33,34,35,36,108]. The identification of potentially repurposable therapeutics for the treatment of inner ear diseases represents a first step towards identifying potentially more efficacious therapies for this disorder and this robust study approach should be considered for other diseases.

### 4.3. Existing Knowledge of Identified Drugs in Hearing Disorders

There has been a lack of drug-discovery research across many of these inner ear diseases (MD, SSNHL, AIED). The drugs identified in this paper were assessed for prior studies investigating their role in the treatment of inner ear diseases and their effect on the auditory system. Some of these drugs will be showcased in this section in research on SSNHL, specifically to demonstrate how further analyses on the drugs identified in this analysis could be studied in inner ear disease. A number of the drugs highlighted below are thought to act synergistically with steroids on anti-inflammatory and neuroprotective pathways [23].

Amitriptyline, a TCA, has been shown to protect and restore synapses and neural function in noise exposed ears in vitro and in vivo in mice, perhaps through glial cell line-derived neurotrophic factor (GDNF) induction [109,110]. Amitriptyline acting as a TrkB agonist is specifically implicated in the regeneration of cochlear synapses [109]. Interestingly, steroids have also been shown to interact with growth factors that act through TrkB receptors [111]. Therefore, amitriptyline may work through growth factors and regulate processes such as cell cycle progression, differentiation, migration and apoptosis, which is thought to be a function of steroid action on the body in pathologic states [112]. Amitriptyline is also used to treat idiopathic tinnitus, vestibular migraine, and facial pain [113,114]. Its neuroprotective effect in these inner-ear and head/neck disorders make it an especially interesting drug for repurposing. Imipramine and doxepin are in the same drug class.

It has been shown that migraine prophylaxis, which includes TCAs and calcium channel blockers, (nortriptyline and verapamil, respectively) in patients with SSNHL may improve hearing outcomes. In a study of 96 patients treated with oral and TT steroids or a combination of steroids and migraine treatment, patients taking the migraine medication had better hearing outcomes [55]. Participants had significantly greater improvements in hearing thresholds at lower frequencies (250 Hz, 500 Hz, and 1000 Hz); and in pure tone averages (PTA) > 10 dB. In another case series of patients with late-stage SSNHL (6 weeks after onset), patients who received prophylactic migraine medication (nortriptyline, topiramate, and/or verapamil) plus lifestyle changes in addition to TTS therapy had improvements in hearing. Of the 21 long-term SSNHL patient cohort, 29% had significant improvement in post-treatment hearing thresholds at multiple frequencies, 68% had improvements in word recognition scores (WRS), and 40% had improvements in speech recognition thresholds (SRT) [56]. TCAs and calcium channel blockers were identified by our druggable analysis and may be involved in cellular processes in SSNHL.

Multiple calcium channel blockers were identified in our analysis, which complements existing data on nimodipine’s use in SSNHL. In a recent retrospective case review of 78 patients diagnosed with SSNHL, one group was treated with steroids and nimodipine and another with steroids only [115]. Although hearing thresholds between the groups did not significantly differ, the complete recovery rate was significantly higher in the nimodipine plus steroid treated patients (41.7% versus 16.8%) and this group had a quicker recovery time (60.9% versus 19.2%) as well. This finding is in contrast to past studies showing that calcium antagonism given alone or in combination with vitamins in patients with SSNHL did not enhance hearing recovery [116,117]. A number of mechanisms shown in animal models are thought to play a role in the improvement of SSNHL such as a protective effect of nimodipine on hair cells, vasodilatory effects which increase cochlear blood flow even in ototoxic conditions, and stabilization of calcium channels which prevents against toxic calcium influx [115,118,119,120]. Therefore, larger-scale clinical trials should be devoted to determining whether calcium blockage exerts a protective effect and improves SSNHL.

The antioxidants identified in our analysis: N-Acetylcysteine (NAC) and zinc have been studied in inner ear diseases. NAC has been shown to be partially protective in noise-induced hearing loss, which can cause sensorineural hearing loss [121,122], and has been shown to ameliorate disease severity in AIED [123,124,125,126]. A number of case–control studies have shown NAC to be effective in SSNHL [127,128]. Hypothesized mechanisms behind this are anti-inflammatory effects, reduction in reactive oxygen species (ROS), and an increase in steroid sensitivity in the inner ear [129,130]. Zinc, through similar anti-inflammatory and neuroprotective mechanisms [131,132,133,134] as NAC, has been hypothesized to be effective in SSNHL in case–control studies [135,136]. Both of these antioxidants may protect hair cells from damage and be therapeutic in inner ear disease.

Existing data supports the notion that the druggable targets focused on in this analysis may be avenues for potentially identifying alternative therapeutic agents for inner ear diseases frequently treated with steroids, including MD, SSNHL, and AIED. The heterogeneity in responses to steroids alone and significant morbidity of these otologic diseases provides a basis to seek additional treatment options when steroid treatment does not result in hearing recovery. These analyses provide a basis for rational drug selection for future repurposing treatment trials for inner ear diseases treated with steroids. More research should be devoted towards understanding the mechanisms of these drugs in the inner ear for a greater understanding of these pathologies.

### 4.4. Limitations

As discussed, the inconsistent results and poorly understood mechanisms with regard to how steroids treat and/or stabilize disease in SSNHL, MD, and AIED makes it challenging to prove the relationship between steroid gene targets identified in the SV and SGN and potential therapeutic benefit of targeting these pathways. Additionally, this study does not address the potential non-genomic effects of steroids, such as effects on membrane excitability that may work independently of genetic alteration [137].

Furthermore, this study only focuses on differential expression of gene targets in the SGN and SV. There are other cell-types within the inner ear, such as single hair cells among others, that merit further investigation. There are currently no published equivalent datasets on the adult stages of these cell types. However, Nelson and colleagues did examine the expression of steroid-responsive gene targets in postnatal day 7 organ of Corti cell types, including supporting cells and hair cells, and identified a limited, albeit distinct set of genes expressed amongst these cell types, although these data are likely not equivalent to adult hair and supporting cells (Appendix A in Nelson et al., 2022) [23]. Given the strong expression of steroid-responsive genes amongst both the SV and SGN, this study focuses on those identified gene targets, providing a foundation for future investigations on these gene targets in other regions of the cochlea. We are not suggesting that these diseases are mediated in the SGN and SV cell types primarily, but these data provide a window into the potential mechanisms that can be studied in these areas.

Another limitation of this study is the use of differential gene expression to infer functional alterations in proteins. While DEGs may provide insights into steroid-responsive phenotypes, it may not correlate with proteomic changes or clinical significance. Furthermore, to localize DEGs to SGN and SV cochlear subtypes, two forms of genetic data were utilized: a bulk gene-expression dataset that tracked gene expression in the whole mouse cochlea based on steroid-administration type, cross referenced with single-cell and single-nucleus datasets to localize the DEGs to these specific cell types [23]. Therefore, the DEG data is limited by the quality of the bulk dataset, and we cannot ensure that cell-type specific expression patterns followed changes in gene expression detected in the whole cochlea. This limits our understanding of the genetic changes within the SGN and SV, and their contribution to inner ear pathologies. Since this DEG analysis is coming from whole cochlea transcriptome analysis, we may not be seeing the whole picture, but we cross-referenced this with known cell type specific databases from single cell transcriptome data, and we utilized a multipronged approach to druggability by using three databases and compiling the results from the three. Thus, future studies should be devoted to mapping genetic and proteomic changes within specific cochlear cell types to elucidate their role in disease states for pharmacologic targeting.

Investigating drug-gene interactions in this study relied upon medications that have been shown to interact with steroid responsive gene targets. This study was unable to determine the functional significance of the degree of gene expression within the cell types or whether greater gene expression may correlate with superior therapeutic efficacy. There are many other genes expressed in the cochlea without differential expression levels, which were not included in our study. Further studies in animal models may aid in elucidating the mechanisms by which alterations in cell type-specific gene expression may relate to inner ear pathologies.

### 4.5. Implications and Future Directions

Through the identification of a list of potentially repurposable FDA-approved therapeutics that target steroid-responsive gene targets, we present a list of potential novel therapeutics for further investigation to treat steroid-responsive inner ear diseases, including SSNHL, MD, and AIED. Furthermore, this study provides a basis to study whether these genes are central to the steroid response in these disorders. Utilization of sequential combinations of these agents may additionally allow for phenotyping of patients by responsiveness to different treatments or combinations thereof. This type of phenotyping may open the door for determining group-specific mechanisms for treatment responsiveness [138]. Deep phenotyping, in the form of genomics and proteomics of both blood and cochlear perilymph, of patients who are both responsive and non-responsive to these treatments may shed light on additional druggable targets and mechanisms responsible for the underlying hearing loss.

There continues to be a lack of proven, effective treatments for many inner ear diseases including SSNHL, MD, and AIED. The lack of effective treatments reflects a general lack of understanding of the underlying mechanisms of these inner ear diseases. Steroids have remained a primary treatment modality due to prior evidence supporting their efficacy, albeit inconsistent. Clinical practice guidelines have supported the use of steroids in many of these disorders [1,2,3]. This study is the first, to our knowledge, to utilize previously described steroid responsive genes in the SGN and SV to investigate potential druggable gene targets. We have provided a foundation for future studies to assess the potential utility of repurposed FDA-approved medications that may interact with these gene targets. Although this study does not provide evidence for the use of these therapies, it proposes a list of potentially novel therapeutics for further evaluation for the treatment of these debilitating otologic disorders.

## Figures and Tables

**Figure 1 biomolecules-12-01641-f001:**
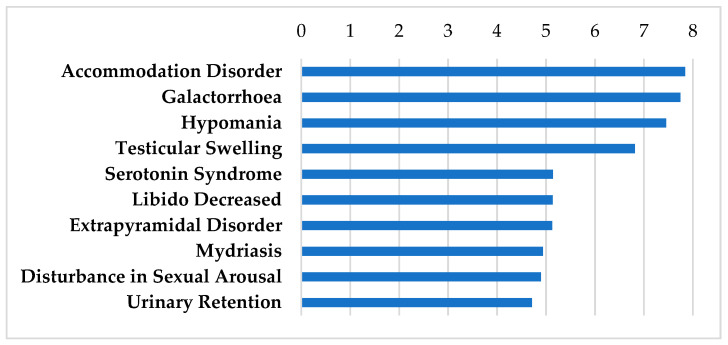
SIDER Side Effect Profile for the antidepressants. These drugs include amitriptyline, doxepin, fluoxetine, fluvoxamine, imipramine, milnacipran, and nefazodone. Frequencies of the top 10 side effects are shown based on the negative logarithm of their respective *p* values.

**Figure 2 biomolecules-12-01641-f002:**
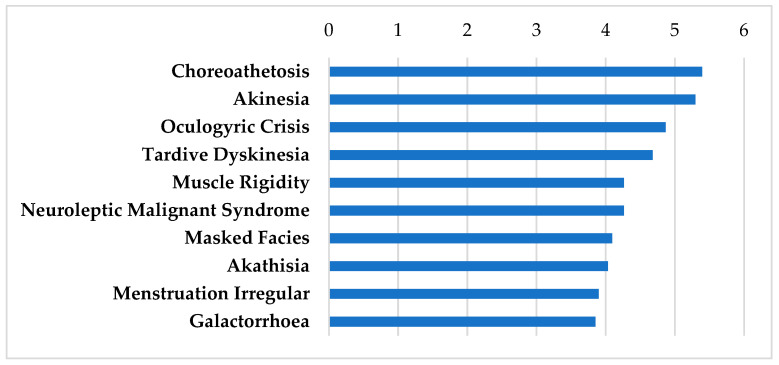
SIDER Side Effect Profile on antipsychotics. These drugs include aripiprazole, thioridazine, and loxapine. Frequencies of the top 10 side effects are shown based on the negative logarithm of their respective *p* values.

**Figure 3 biomolecules-12-01641-f003:**
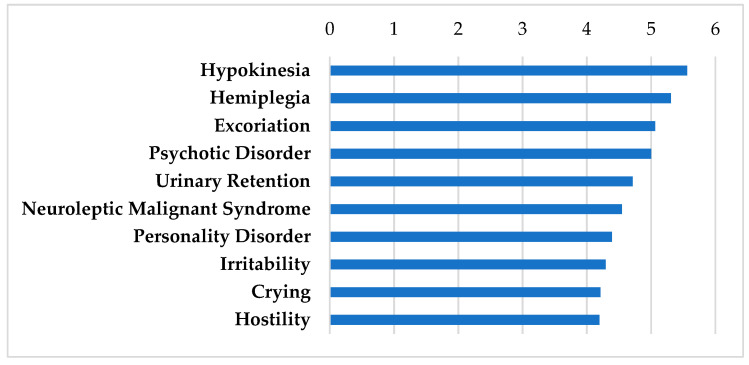
SIDER Side Effect Profile of drugs that work in the central nervous system. These drugs include donepezil, gabapentin, memantine, phenytoin, amantadine, and atomoxetine. Frequencies of the top 10 side effects shown are displayed according to the negative logarithm of their respective *p* values.

**Figure 4 biomolecules-12-01641-f004:**
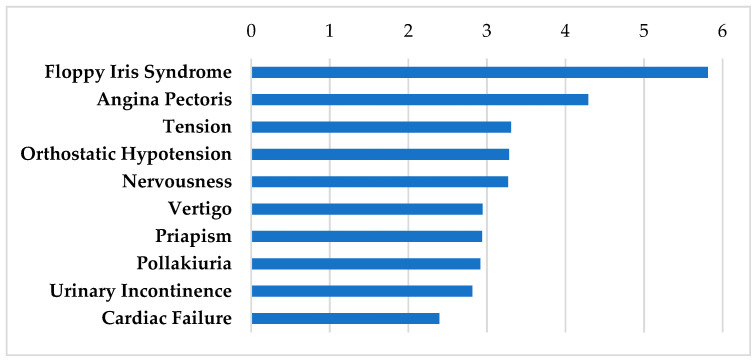
SIDER side effect profile on adrenergic modulators. These drugs include carvedilol, doxazosin, prazosin, terazosin, ephedrine, and tizanidine. Frequencies of the top 10 side effects are shown based on the negative logarithm of their respective *p* values.

**Figure 5 biomolecules-12-01641-f005:**
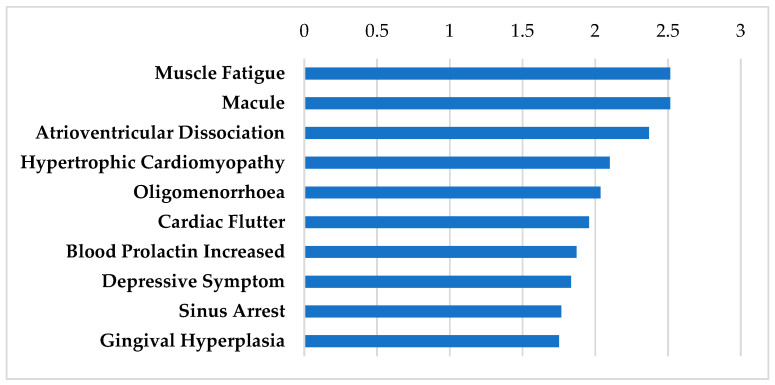
SIDER side effect profile on calcium channel blockers. Frequencies of the top 10 side effects shown for nifedipine and verapamil are based on the negative logarithm of their respective *p* values.

**Figure 6 biomolecules-12-01641-f006:**
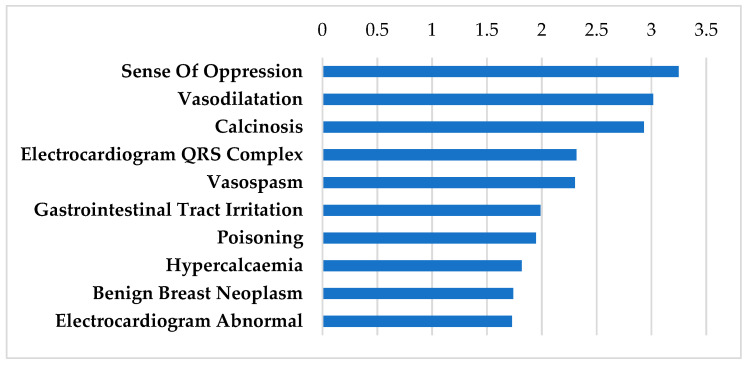
SIDER side effect profile on ion modulators including calcium supplementation and spironolactone. Frequencies of the top 10 side effects shown for the following drugs: calcium citrate, calcium phosphate, and spironolactone based on the negative logarithm of their respective *p* values.

## Data Availability

Lists of the druggable differentially expressed genes are available in Appendix A. The data utilized in this study regarding identification and lists of steroid-responsive genes in the mammalian spiral ganglion and stria vascularis subtypes are previously published by Nelson et al., 2022 at 10.3389/fneur.2021.818157 [23] in Appendix A. Further inquiries can be directed to the corresponding author.

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
