# Peer review of "Repurposable Drugs That Interact with Steroid Responsive Gene Targets for Inner Ear Disease"

_biomolecules, 2022, doi:10.3390/biom12111641_

Round 1

Reviewer 1 Report

Title: Repurposable drugs that interact with steroid responsive gene targets in sudden sensorineural hearing loss

This is clearly written and well-organized paper detailing a brute force methodology to catalog 48 inner ear genes in inner ear cells that were significantly up- or down-regulated by steroid administration.  That aspect of the paper is solid.  The title of the study, the rationale for performing the study, and the attempt in the Discussion to argue that the findings have any direct relevance to sudden sensorineural hearing loss (SSNHL), however, is unjustified.  The mechanism of SSNHL is absolutely unknown.  Studies of post mortem human temporal bones have failed to show evidence of inflammation or ischemia.  The multiple clinical trials of oral and/or intratympanic corticosteroids, only one of which ever met its accrual target (Rauch et al., JAMA 2011), have never shown convincing evidence of steroid efficacy.  In fact, the most recent Cochrane Review on this topic concludes that steroids have never been shown to have better efficacy than placebo.  Considering (i) the lack of evidence that inflammation or immune dysregulation play any role in SSNHL, (ii) the lack of any animal model for this condition, and (iii) the demonstrated extreme logistical difficulty of performing a placebo controlled RCT of SSNHL in humans, would suggest that any claim by the authors of the current manuscript that their findings advance the field of SSNHL management is overreach.  Which of the 48 putative targets would they recommend studying?  And how?  It would be much more plausible and appropriate to retitle the paper by changing the last four words, “sudden sensorineural hearing loss”, to “inner ear disease”.  The Introduction and Discussion can then be rewritten to highlight how these identified “druggable targets” may be of theoretical utility IF anyone ever discovers an inner ear disease mediated by one of those targets.  Even though the scientific work done by the authors was well-executed and well-described in the manuscript, hitching this project to the specific condition of SSNHL “wagon” is not appropriate.

Reviewer 2 Report

The manuscript uses previously identified DEGs upon steroid-responses in SV and SGN to compare to gene target databases for repurposing FDA-approved drugs. This approach is relatively new. They further identify 42 FDA-approved drugs interacting with these molecular targets with good safety and tolerability profiles. These drug candidates are basis for treatment of SSNHL. In particular, the use of SIDER and Drugmonizome is powerful in identifying the safety profiles and tolerability. The manuscript is well written. I recommend revision for publication.

1.       The main concern is that nothing is new, even the use of SIDER and Drugmonizome is not novel. Please emphasize this fact in the text (limitations).

2.       Everything they are presenting is circumstantial. They wrote a short section about the limitations of the study and they did that on purpose so it won’t draw much attention and because they needed to include it. They are assuming that the beneficial effects of steroids are just coming from the SV and the SGN, because it is where the major DEG were found. That is pure speculation. Also, they don’t say anything about the non-genomic effects of steroids, especially corticosterone. It has been known for some time now that steroids also interact with membrane receptors and by doing that, they can trigger rapid responses. The authors kind of mention that in the limitation section but it is very vague.

3.       They want to show that their method of processing and analysis of the data is correct because in the final outcomes they obtained compounds such as NAC, that is already known it can protect hearing. But I think is because it is kind skewed towards that. They have a small pool of DEG genes and are trying to compare them with DEGs that are coming mainly from cancer cells, so they are always going to end up with the same kind of drugs when using LINCs.

4.       Please indicate that no studies are performed on sensory hair and supporting cells which may play equally important roles in SSNHL. In fact, some drugs may not interact with targets in SV or SNG but in organ of Corti cells instead (line 130 and see comment #2 abiove).

5.       Specify drugs excluded from the US market (line 137).

Reviewer 3 Report

In general, the article provides other drug choices for patients with sudden deafness who are not sensitive to steroids and has obtained positive results in some drugs. However, the analysis of the data in this paper is relatively simple and lacks an interpretation of the results and an explanation of the pharmacological and pathophysiological mechanisms in both positive and negative cases.

1.     Further details are needed in the methods to describe the number and age of mice administered as well as the duration and amount of administration in the systemic steroid (SS) group, the transseptal steroid (TTS) group, and the saline group, the authors cite previous data that need to be clearly stated in the methods. Is there a difference between the results of this study and those of Nelson et al. using the same transcriptomic data cross-over analysis with single-cell data that has been published (PMID: 35145472)? The authors need to show the results of the cross-tabulation analysis first.

2.     The authors' previous study found that the genes affected in the cochlea varied greatly between different steroids and different modes of administration. In this paper, they need to specify which steroids were used and what were the differences and similarities in the findings when different steroids were analyzed in association with drugs. Do the conclusions of this study change when third-party data (PMID: 28306650) are used?

3.     How to explain the effect of Zinc acetate on Tufm in Table 4. The result of SGN: TTS>Control is NC, while that of SGN: SS>Control is Y.

4.     What is the purpose of listing the side effects of drugs? Is there any relationship between the side effects of the drug and its therapeutic effect on sudden deafness?  How to define the application of drugs?

5.     The conclusions shown in the title cannot be drawn through the findings of this study, and the title is suggested to be changed to a more accurate description.

Round 2

Reviewer 1 Report

The authors have satisfactorily addressed the reviewer's concerns.

Reviewer 3 Report

The author answered my questions very well and made changes to the manuscript.